# Approximate Differential Privacy of the $\ell_2$ Mechanism

**Matthew Joseph** [* 1]   **Alex Kulesza** [* 1]   **Alexander Yu** [* 1]

## Abstract

We study the $\ell_2$ mechanism for computing a $d$-dimensional statistic with bounded $\ell_2$ sensitivity under approximate differential privacy. Across a range of privacy parameters, we find that the $\ell_2$ mechanism obtains lower error than the Laplace and Gaussian mechanisms, matching the former at $d = 1$ and approaching the latter as $d \to \infty$.

## 1. Introduction

Computing a $d$-dimensional statistic with bounded $\ell_2$ sensitivity is a fundamental task in differential privacy (DP) (Dwork et al., 2006). It underlies standard algorithms like private stochastic gradient descent (Song et al., 2013; Abadi et al., 2016), the binary tree mechanism (Chan et al., 2011; Dwork et al., 2010), and the projection (Nikolov et al., 2013; Nikolov, 2023), matrix (Li et al., 2015; McKenna et al., 2018), and factorization mechanisms (Edmonds et al., 2020; Nikolov & Tang, 2023). The canonical approximate DP algorithm for this problem is the Gaussian mechanism (Dwork et al., 2006). To compute statistic $T(X)$, the Gaussian mechanism samples an output according to $g_X(y) \propto \exp(-[\|y - T(X)\|_2/\sigma]^2)$ for an appropriate value of $\sigma$; in particular, the analytic Gaussian mechanism (Balle & Wang, 2018) chooses the smallest possible $\sigma$ sufficient for the desired approximate DP guarantee.

In this paper, we analyze the $\ell_2$ mechanism. Given a parameter $\sigma$, this mechanism samples an output according to density $f_X(y) \propto \exp(-\|y - T(X)\|_2/\sigma)$. As an instance of the $K$-norm mechanism (Hardt & Talwar, 2010) using the $\ell_2$ norm, the $\ell_2$ mechanism immediately satisfies $\frac{1}{\sigma}$-(pure) DP and can be sampled efficiently. However, its approximate DP guarantees are not well understood.

*Equal contribution [1]Google Research New York. Correspondence to: Matthew Joseph <mtjoseph@google.com>.

*Proceedings of the $42^{nd}$ International Conference on Machine Learning*, Vancouver, Canada. PMLR 267, 2025. Copyright 2025 by the author(s).

### 1.1. Contributions

For arbitrary dimension $d$ and privacy parameters $\varepsilon$ and $\delta$, we provide an algorithm for choosing $\sigma$ to obtain an $\ell_2$ mechanism that satisfies $(\varepsilon, \delta)$-DP. The resulting $\ell_2$ mechanism can be efficiently sampled in parallel and empirically dominates both the Laplace mechanism and the analytic Gaussian mechanism in terms of mean squared $\ell_2$ error (left plot in Figure 1). Moreover, unlike the Gaussian mechanism, the $\ell_2$ mechanism always satisfies a pure DP guarantee (right plot in Figure 1).

Our algorithms bound relevant quantities of the privacy loss random variable for the $\ell_2$ mechanism. Balle & Wang (2018) showed that mechanism $M$ is $(\varepsilon, \delta)$-DP if and only if

$$\mathbb{P}\left[\ell_{M,X,X'} \geq \varepsilon\right] - e^{\varepsilon}\mathbb{P}\left[\ell_{M,X',X} \leq -\varepsilon\right] \leq \delta \quad (1)$$

where $\ell_{M,X,X'}$ is the privacy loss associated with $M$ on arbitrary neighboring databases $X$ and $X'$ (Section 2). Proving $(\varepsilon, \delta)$-DP therefore reduces to upper bounding the first term and lower bounding the second. We show that the first term is defined by the mass that $M(X)$ places on a region of $\mathbb{R}^d$ determined by certain spherical caps, while the second term is defined by the mass that $M(X')$ places on the same region. We then provide algorithms to approximate the first term from above and the second term from below. Because these approximations are provably upper and lower bounds, they yield a formal differential privacy guarantee. Experiments suggest that, for reasonable algorithm parameter values, these approximations are tight (Section 4.1).

### 1.2. Related Work

Ganesh & Zhao (2021) also use spherical caps to analyze what they call "generalized Gaussians", which have densities proportional to $\exp(-[\|y - T(X)\|_p/\sigma]^p)$ for integers $p \geq 1$. A few features separate their work from ours: they study a statistic with bounded $\ell_\infty$ sensitivity; their results do not cover the $\ell_2$ mechanism, which uses norm $p = 2$ but exponent $p' = 1$; they work with a sufficient condition for $(\varepsilon, \delta)$-DP, which only bounds the first term in Equation (1) by $\delta$, leading to looser results; and since their goal is an asymptotic utility guarantee, their results rely on asymptotic concentration inequalities that are less precise than the approach used here.

A few authors have studied the $\ell_2$ mechanism, primarily in

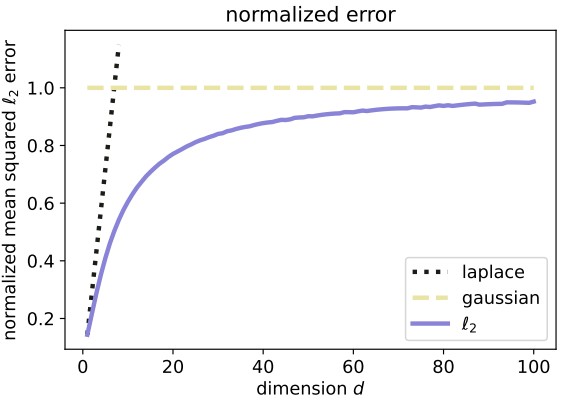
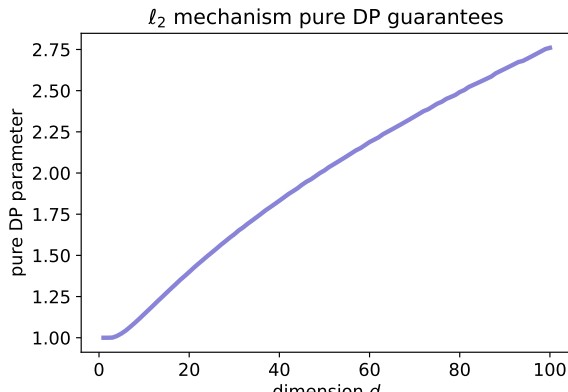

*Figure 1.* **Left**: normalized mean squared $\ell_2$ error. At each $d$, we compute mean squared $\ell_2$ error for the $(1, 10^{-5})$-DP Laplace, analytic Gaussian (Balle & Wang, 2018), and $\ell_2$ mechanisms. Quantities are normalized so that the analytic Gaussian mechanism error is always 1. Note that we truncate the Laplace mechanism at $d = 8$, after which its error relative to the analytic Gaussian mechanism continues to grow. See Section 4.2 for details. **Right**: the pure DP guarantee of the $(1, 10^{-5})$-DP $\ell_2$ mechanism as $d$ grows.

the context of objective perturbation (Chaudhuri et al., 2011; Kifer et al., 2012; Yu et al., 2014). However, they all use pure DP rather than approximate DP.

## 2. Preliminaries

We use the formulation of $(\varepsilon, \delta)$-DP given by Balle & Wang (2018). It is defined in terms of the privacy loss random variable. Our results apply for either the add-remove or swap notions of neighboring databases.

**Definition 2.1.** *Let $M$ be a mechanism whose output density given input database $X$ is $f_X$. Then for neighboring databases $X, X'$, the privacy loss of $M$ at point $y$ is $\ell_{M,X,X'}(y) = \ln\left(\frac{f_X(y)}{f_{X'}(y)}\right)$. Its privacy loss random variable is $\ell_{M,X,X'}(Y)$ where $Y \sim f_X$.*

The following results relate the privacy loss random variable to differential privacy.

**Lemma 2.2.** *Mechanism $M$ is $\varepsilon$-DP if and only if, for any neighboring $X \sim X'$, $|\ell_{M,X,X'}(Y)| \leq \varepsilon$.*

**Lemma 2.3** (Balle & Wang (2018))**.** *Mechanism $M$ is $(\varepsilon, \delta)$-DP if and only if, for any neighboring $X \sim X'$,*

$$\mathbb{P}\left[\ell_{M,X,X'} \geq \varepsilon\right] - e^\varepsilon \mathbb{P}\left[\ell_{M,X',X} \leq -\varepsilon\right] \leq \delta.$$

Since the $\ell_2$ mechanism can be viewed as an instance of the $K$-norm mechanism (Hardt & Talwar, 2010), we recall some relevant results about the $K$-norm mechanism.

**Lemma 2.4** (Hardt & Talwar (2010))**.** *Given norm $\|\cdot\|$, scale parameter $\sigma$, statistic $T$ with $\|\cdot\|$-sensitivity $\Delta$, and database $X$, the $K$-norm mechanism has output density $f_X(y) \propto \exp\left(-\|y - T(X)\|/\sigma\right)$ and satisfies $\frac{\Delta}{\sigma}$-DP. Moreover, letting $B^d$ denote the unit ball for $\|\cdot\|$, the following procedure samples this mechanism: 1) sample radius*

$r \sim \mathsf{Gamma}\,(d+1, \sigma)$, *the Gamma distribution with shape $d + 1$ and scale $\sigma$; 2) uniformly sample $z \sim B^d$; and 3) output $T(X) + rz$.*

## 3. $\ell_2$ Mechanism

This section provides an algorithm for computing $\sigma$ to achieve an $(\varepsilon, \delta)$-DP $\ell_2$ mechanism (Section 3.1) and then describes a simple method for sampling the $\ell_2$ mechanism in parallel (Section 3.2). Without loss of generality, we assume that our statistic $T$ has $\ell_2$ sensitivity $\Delta_2 = 1$. If $\Delta_2 \neq 1$, we can run the algorithm on $T/\Delta_2$ and rescale.

### 3.1. Privacy Analysis

The overall goal is to translate an $(\varepsilon, \delta)$-DP privacy budget to the minimum $\sigma$ such that the $\ell_2$ mechanism $M$ with parameter $\sigma$ satisfies $(\varepsilon, \delta)$-DP. To do this, we focus on a subroutine that determines whether or not $M$ satisfies $(\varepsilon, \delta)$-DP and then binary search over $\sigma$.

Recall from Lemma 2.3 that $M$ is $(\varepsilon, \delta)$-DP if and only if $\mathbb{P}\left[\ell_{M,X,X'} \geq \varepsilon\right] - e^\varepsilon \mathbb{P}\left[\ell_{M,X',X} \leq -\varepsilon\right] \leq \delta$. The next subsections will provide algorithms that upper bound the first term and lower bound the second term, and thus err on the side of a conservative privacy guarantee.

Before starting our privacy analysis, we consider a simpler (but, as we will see, significantly worse) approach. The $\ell_2$ mechanism, and the $K$-norm mechanism more broadly, can be viewed as instances of the exponential mechanism (McSherry & Talwar, 2007). The exponential mechanism admits a few possible approximate DP analyses. For example, the $\varepsilon$-DP exponential mechanism satisfies $\frac{\varepsilon^2}{8}$-concentrated DP (Cesar & Rogers, 2021), and a concentrated DP guarantee can be converted to an approximate DP guarantee (Bun

& Steinke, 2016; Canonne et al., 2020; Asoodeh et al., 2020; Zhu et al., 2022). However, any such analysis also applies to the Laplace mechanism, and the $\varepsilon$-DP Laplace mechanism is only $(\varepsilon', \delta)$-DP for $\varepsilon' \approx \varepsilon - O(\delta)$ (Lemma A.2 in the Appendix). This is a negligible improvement for realistic $\delta$, so a different privacy analysis is necessary.

### 3.1.1. FIRST TERM UPPER BOUND

For the privacy guarantee to hold, Equation (1) must hold for arbitrary neighboring databases $X$ and $X'$. Since the $\ell_2$ ball is spherically symmetric, without loss of generality we consider statistic $T$ where $T(X) = 0$ and $T(X') = e_1 = (1, 0, \dots, 0)$. Shorthand the respective mechanisms as $M(0)$ and $M(1)$. Then

$$\begin{aligned} \ell_{M,X,X'}(y) &= \ln\left(\frac{f_X(y)}{f_{X'}(y)}\right) \\ &= \ln\left(\frac{\exp[-\|y\|_2/\sigma]}{\exp[-\|y - e_1\|_2/\sigma]}\right) \\ &= \frac{1}{\sigma}\left(\|y - e_1\|_2 - \|y\|_2\right) \end{aligned}$$

so we want to upper bound

$$\mathbb{P}_{y\sim M(0)}\left[\frac{1}{\sigma}\left(\|y - e_1\|_2 - \|y\|_2\right) \geq \varepsilon\right]. \quad (2)$$

We shorthand the relevant region in Equation (2) as $V$.

**Definition 3.1.** *Define* $V = \{y \mid \frac{1}{\sigma}(\|y - e_1\|_2 - \|y\|_2) \geq \varepsilon\}$, $M$'s high privacy loss region.

A simple case is $\sigma \geq \frac{1}{\varepsilon}$.

**Lemma 3.2.** $\sigma \geq \frac{1}{\varepsilon}$ *if and only if* $\mathbb{P}_{y\sim M(0)}[y \in V] = 0$.

*Proof.* The equation $|\|y - e_1\|_2 - \|y\|_2| = \sigma\varepsilon$ defines a hyperboloid with foci $0$ and $e_1$ and constant difference $\sigma\varepsilon$. If we instead consider $\|y - e_1\|_2 - \|y\|_2 \geq \sigma\varepsilon$, removing the absolute value restricts the hyperboloid to the $-e_1$ facing component, and moving to inequality yields the convex hull of that component. An illustration appears in Figure 2.

If $\sigma > \frac{1}{\varepsilon}$, then $\|y - e_1\|_2 - \|y\|_2 \geq \sigma\varepsilon$ has no solution because the triangle inequality means $\|y - e_1\|_2 - \|y\|_2 \leq \|e_1\|$. Thus $V = \emptyset$, so $\mathbb{P}_{y\sim M(0)}[y \in V] = 0$. If $\sigma = \frac{1}{\varepsilon}$, then $\|y - e_1\|_2 - \|y\|_2 \geq \sigma\varepsilon$ only holds for $y$ contained in the $-e_1$ axis. This set has measure 0, so $\mathbb{P}_{y\sim M(0)}[y \in V] = 0$. Finally, if $\sigma < \frac{1}{\varepsilon}$, then $\|y - e_1\|_2 - \|y\|_2 \geq \sigma\varepsilon$ determines a $-e_1$ facing component of the hyperboloid that is non-degenerate, so its convex hull has positive measure, i.e. $\mathbb{P}_{y\sim M(0)}[y \in V] > 0$. $\square$

The rest of this subsection considers $\sigma < \frac{1}{\varepsilon}$. When $d = 1$, all $\ell_p$ norm mechanisms are identical. In particular, the $\ell_2$ mechanism is equivalent to the Laplace mechanism.

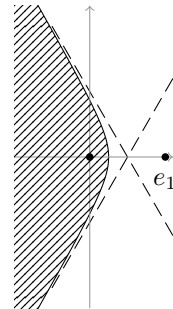

Figure 2. An illustration of $V$ for $\sigma = 1/(2\varepsilon)$. We draw the projection of $V$ onto $\mathrm{span}(e_1, e_2)$ as the shaded region.

**Lemma 3.3.** *If* $\sigma \leq \frac{1}{\varepsilon}$ *and* $d = 1$, *then* $\mathbb{P}_{y\sim M(0)}[y \in V] = 1 - \frac{1}{2}\exp\left(\frac{1}{2}[\varepsilon - \frac{1}{\sigma}]\right)$.

*Proof.* $M$ has the same noise density as the Laplace mechanism $\mathsf{Lap}(\sigma)$, and $|y - e_1| - |y| \geq \sigma\varepsilon$ if and only if $y \leq \frac{1}{2}(1 - \sigma\varepsilon)$. For $z \geq 0$, the $\mathsf{Lap}(\sigma)$ CDF is $F(z) = 1 - \frac{1}{2}\exp\left(-\frac{z}{\sigma}\right)$, so by $1 - \sigma\varepsilon \geq 0$, $\mathbb{P}_{y\sim M(0)}[y \in V] = 1 - \frac{1}{2}\exp\left(\frac{1}{2}[\varepsilon - \frac{1}{\sigma}]\right)$. $\square$

This leaves the case $\sigma \leq \frac{1}{\varepsilon}$ and $d \geq 2$. We proceed under that assumption.

**Assumption 3.4.** *Statistic* $T$ *has dimension* $d \geq 2$, *and* $\sigma < \frac{1}{\varepsilon}$.

When $d \geq 2$, the level sets of $M$ are spheres, and we will show that the the high privacy loss portions of level sets are spherical caps.

**Definition 3.5.** *For* $r > 0$ *and* $z \in \mathbb{R}^d$, *define* $S_{r,z} = \{x \in \mathbb{R}^d \mid \|x - z\|_2 = r\}$, *the sphere of radius* $r$ *centered at* $z$. *For any sphere* $S_{r,z}$ *where* $z = (c, 0, \dots, 0)$, *the spherical cap of* $S_{r,z}$ *of height* $h$ *is the set of points* $\hat{S}_{r,z,h} = \{(x_1, \dots, x_d) \in \mathbb{R}^d \mid \|x - z\|_2 = r \text{ and } x_1 \leq c - r + h\}$. *See Figure 3 for an illustration.*

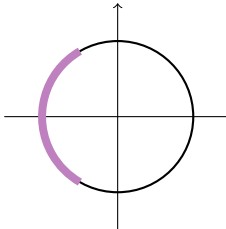

Figure 3. The unit circle in $\mathbb{R}^2$ with a spherical cap (thick purple arc) of height 0.5. In $\mathbb{R}^d$, the sphere and spherical cap are both $(d-1)$-dimensional objects.

**Lemma 3.6.** *Define height*

$$h(r) = \min\left(r(1 - \varepsilon\sigma) + \frac{1 - (\varepsilon\sigma)^2}{2}, 2r\right).$$

*Then $\hat{S}_{r,0,h(r)} = V \cap S_{r,0}$.*

*Proof.* Orient the 2-dim plane $\text{span}(e_1, e_2)$ so that the positive $e_1$ direction is right and the positive $e_2$ direction is up. Consider the points $0$, $e_1$, and some $y \in S_{r,0} \cap H$ where $H$ is the upper half of the $\text{span}(e_1, e_2)$ plane (i.e., $y_2 \geq 0$). Let $\theta$ be the clockwise angle from $y$ to $e_1$. Proofs of the following claim (and others omitted in this subsection) appear in Appendix A.1.

**Claim 3.6.1.** $\|y - e_1\|_2 - \|y\|_2$ *decreases as $\theta$ decreases.*

We want to identify a function $h(r)$ for all $r > 0$ such that $\hat{S}_{r,0,h(r)} = V \cap S_{r,0}$. Assumption 3.4 means $\sigma < \frac{1}{\varepsilon}$, so by Lemma 3.2, $V \cap S_{r,0}$ is nonempty. The analysis splits into cases.

Case 1: $S_{r,0} \not\subset V$. Since $\frac{1}{\sigma}(\|p - e_1\|_2 - \|p\|_2)$ changes monotonically by Claim 3.6.1 then there exists $p = (p_1, p_2, 0, ..., 0) \in S_{r,0}$ at the base of $\hat{S}_{r,0,h(r)}$ such that $\frac{1}{\sigma}(\|p - e_1\|_2 - \|p\|_2) = \varepsilon$. Then $h(r)$ satisfies $(r - h(r))^2 + p_2^2 = r^2$, and we get $p_2 = \sqrt{2h(r)r - h(r)^2}$.

We can now derive the desired expression for $h(r)$.

**Claim 3.6.2.** $\frac{1}{\sigma}(\|p - e_1\|_2 - \|p\|_2) = \varepsilon$ *is equivalent to* $h(r) = r(1 - \varepsilon\sigma) + \frac{1 - \varepsilon^2\sigma^2}{2}$.

Moreover, we show that with the above definition of $h(r)$, the constraint $h(r) \in [0, 2r]$ is equivalent to a constraint on the radius given by $r \geq \frac{1 - \varepsilon\sigma}{2}$.

**Claim 3.6.3.** $r(1 - \varepsilon\sigma) + \frac{1 - \varepsilon^2\sigma^2}{2} \in [0, 2r]$ *if and only if* $r \geq \frac{1 - \varepsilon\sigma}{2}$.

In summary, $S_{r,0} \not\subset V$ if and only if $r \geq \frac{1 - \varepsilon\sigma}{2}$, and for such $r$, we have $\hat{S}_{r,0,h(r)} = V \cap S_{r,0}$ when $h(r)$ is defined as in Claim 3.6.2.

Case 2: $S_{r,0} \subset V$. By the above, $S_{r,0} \subset V$ if and only if $r < \frac{1 - \varepsilon\sigma}{2}$. The statement $S_{r,0} \subset V$ is equivalent to $\frac{1}{\sigma}(\|p - e_1\|_2 - \|p\|_2) = \frac{1}{\sigma}(\sqrt{(r+1)^2 - 2h(r)} - r) \geq \varepsilon$ for all $p \in S_{r,0}$, and this inequality is equivalent to $h(r) \leq r(1 - \varepsilon\sigma) + \frac{1 - \varepsilon^2\sigma^2}{2}$ by replacing equality with inequalities in the proof of Claim 3.6.2. By Claim 3.6.3, $r < \frac{1 - \varepsilon\sigma}{2}$ is equivalent to $r(1 - \varepsilon\sigma) + \frac{1 - \varepsilon^2\sigma^2}{2} > 2r$. Then defining $h(r) = 2r$ suffices to satisfy $h(r) \leq r(1 - \varepsilon\sigma) + \frac{1 - \varepsilon^2\sigma^2}{2}$ for all $r < \frac{1 - \varepsilon\sigma}{2}$.

In summary, if we define $h(r)$ as

$$h(r) = \begin{cases} r(1 - \varepsilon\sigma) + \frac{1 - \varepsilon^2\sigma^2}{2} & \text{if } r \geq \frac{1 - \varepsilon\sigma}{2} \\ 2r & \text{if } r < \frac{1 - \varepsilon\sigma}{2} \end{cases}$$

then $h(r) \in [0, 2r]$ and $\hat{S}_{r,0,h(r)} = V \cap S_{r,0}$. Since $r(1 - \varepsilon\sigma) + \frac{1 - \varepsilon^2\sigma^2}{2} > 2r$ for $r < \frac{1 - \varepsilon\sigma}{2}$, this piecewise function is equivalent to $h(r) = \min\left(r(1 - \varepsilon\sigma) + \frac{1 - (\varepsilon\sigma)^2}{2}, 2r\right)$. $\square$

We showed in the above proof that, for small $r$, the entirety of $S_{r,0}$ lies in $V$.

**Corollary 3.7.** *If $r \leq \frac{1 - \varepsilon\sigma}{2}$, then $S_{r,0} \subset V$.*

It remains to analyze the high privacy loss region for larger $r$. Our analysis will repeatedly reason about the fraction of a sphere occupied by a cap.

**Definition 3.8.** *Let $F_{r,h}$ denote the fraction of the surface of $S_{r,0}$ occupied by cap $\hat{S}_{r,0,h}$.*

**Lemma 3.9** ((Li, 2010)). *Let $I_x(a, b) = \frac{\int_0^x t^{a-1}(1-t)^{b-1} dt}{\int_0^1 t^{a-1}(1-t)^{b-1} dt}$ denote the regularized incomplete beta function. Let $h$ be the height function defined in Lemma 3.6. If $h(r) \leq r$, then $F_{r,h(r)} = \frac{1}{2} I_{(2rh(r) - h(r)^2)/r^2}\left(\frac{d-1}{2}, \frac{1}{2}\right)$. If $h(r) > r$, then $F_{r,h(r)} = 1 - F_{r,2r-h(r)}$.*

There is no closed-form expression for $I_x(a, b)$, but it is a standard function in mathematical libraries like SciPy (SciPy, 2024).[1] We can therefore use Lemma 3.9 to compute the fraction of any $S_{r,0}$ that lies in $V = \cup_{r \in \mathbb{R}^+} \hat{S}_{r,0,h(r)}$ (Lemma 3.6). It remains to extend these results about individual spheres to results about $V$ as a whole.

The next lemma shows that the high-loss cap fraction decreases with $r$. It combines Lemma 3.6 and Lemma 3.9 to show directly that the appropriate $I_x(a, b)$ is monotone in $r$ in the desired direction. Since the proof is again mostly calculation, it also appears in Appendix A.1.

**Lemma 3.10.** *$F_{r,h(r)}$ is monotone decreasing in $r$.*

This suggests the following approach: if $M(0)$ samples $y$ with $\|y\|_2 > r$ with probability $p$, then $pF_{r,h(r)}$ is an upper bound on $\mathbb{P}_{y \sim M(0)}[y \in V, \|y\|_2 > r]$. The following result gives a closed form for $p$. The expression uses the lower incomplete gamma function and the gamma function, which are also standard functions in mathematical libraries. The proof is mostly calculation and appears in Appendix A.1.

**Definition 3.11.** *For $z \geq 0$, the Gamma function is $\Gamma(z) = \int_0^\infty t^{z-1} e^{-t} dt$, the lower incomplete Gamma function is $\gamma(z, x) = \int_0^x t^{z-1} e^{-t} dt$, and the upper incomplete Gamma function is $\bar{\Gamma}(z, x) = \Gamma(z) - \gamma(z, x)$.*

**Lemma 3.12.** *For $r > 0$, $\mathbb{P}_{y \sim M(0)}[\|y\|_2 \leq r] = \frac{\gamma(d, r/\sigma)}{\Gamma(d)}$.*

The preceding results yield the following upper bound.

---

[1]Note that the analytic Gaussian mechanism (Balle & Wang, 2018) depends similarly on the standard Gaussian CDF.

**Lemma 3.13.** *Suppose Assumption 3.4 holds. Let $r_1 < \ldots < r_{n_r}$ where $r_1 = \frac{1-\varepsilon\sigma}{2}$. Then, for $y \sim M(0)$,*

$$
\begin{aligned}
\mathbb{P}\left[y \in V\right] &\leq \frac{\gamma(d, r_1/\sigma)}{\Gamma(d)} \\
&+ \sum_{j=1}^{n_r-1} \frac{\gamma(d, r_{j+1}/\sigma) - \gamma(d, r_j/\sigma)}{\Gamma(d)} F_{r_j, h(r_j)} \\
&+ \frac{\Gamma(d, r_{n_r}/\sigma)}{\Gamma(d)} F_{r_{n_r}, h(r_{n_r})}.
\end{aligned}
$$

*Proof.* The first term is the mass placed on the ball lying entirely in the high privacy loss region (Corollary 3.7 and Lemma 3.12); the second term is a (left) Riemann sum that upper bounds the mass placed on the high privacy loss region between balls of radius $r_1$ and $r_{n_r}$ (Lemma 3.9, Lemma 3.12, and Lemma 3.10); and the last is an upper bound on the mass placed on the high privacy loss region outside the ball of radius $r_{n_r}$. More specifically, it is a Riemann sum in which the $j$th approximating rectangle has base length as the $j$th interval on the grid $\left[0, \frac{\gamma(d, r_1/\sigma)}{\Gamma(d)}, \ldots, \frac{\gamma(d, R_{n_r}/\sigma)}{\Gamma(d)}, \infty\right]$ and height $F_{r_j, h(r_j)}$. $\square$

Algorithm 1 provides overall upper bound pseudocode.

---

**Algorithm 1** Term1UpperBound

---

1: **Input:** Dimension $d$; scale parameter $\sigma$; privacy parameter $\varepsilon$; largest radius $r^*$; number of radii $n_r$
2: **if** $\sigma > 1/\varepsilon$ **then**
3:     Return 0 (Lemma 3.2)
4: **end if**
5: **if** $d = 1$ **then**
6:     Return $1 - \frac{1}{2}\exp\left(\frac{1}{2}\left[\varepsilon - \frac{1}{\sigma}\right]\right)$ (Lemma 3.3)
7: **end if**
8: Define $r_1 \leftarrow \frac{1-\varepsilon\sigma}{2}$, $r_{n_r} \leftarrow r^*$, and $r_2, \ldots, r_{n_r-1}$ regularly spaced between $r_1$ and $r_{n_r}$
9: Return upper bound from Lemma 3.13

---

### 3.1.2. SECOND TERM LOWER BOUND

It remains to lower bound $\mathbb{P}_{y \sim M(1)}\left[\ell_{M,X',X}(y) \leq -\varepsilon\right]$. By the same logic used to derive Equation (2), we rewrite it as

$$
\begin{aligned}
&\mathbb{P}_{y \sim M(1)}\left[\ln\left(\frac{f'_X(y)}{f_X(y)}\right) \leq -\varepsilon\right] \\
&= \mathbb{P}_{y \sim M(1)}\left[\frac{1}{\sigma}\left(\|y - e_1\|_2 - \|y\|_2\right) \geq \varepsilon\right].
\end{aligned}
$$

The level sets of $M(1)$ are spheres centered at $e_1$. As in the upper bound analysis, there are a few simple cases. The first follows from the same reasoning used for Lemma 3.2.

**Corollary 3.14.** *If $\sigma \geq \frac{1}{\varepsilon}$, then $\mathbb{P}_{y \sim M(1)}\left[y \in V\right] = 0$.*

The second case uses the same argument as Lemma 3.3.

**Lemma 3.15.** *If $\sigma \leq \frac{1}{\varepsilon}$ and $d = 1$, then $\mathbb{P}_{y \sim M(1)}\left[y \in V\right] = \frac{1}{2}\exp\left(\frac{1}{2}\left[-\varepsilon - \frac{1}{\sigma}\right]\right)$.*

*Proof.* $M(1)$ has the same noise density as the Laplace mechanism $\mathsf{Lap}\left(1, \sigma\right)$, and $|y - e_1| - |y| \geq \sigma\varepsilon$ if and only if $y \leq \frac{1}{2}(1 - \sigma\varepsilon)$. For $z < 1$, the $\mathsf{Lap}\left(1, \sigma\right)$ CDF is $F(z) = \frac{1}{2}\exp\left(\frac{z-1}{\sigma}\right)$, so by $1 - \sigma\varepsilon \geq 0$, $\mathbb{P}_{y \sim M(1)}\left[y \in V\right] = \frac{1}{2}\exp\left(\frac{1}{2}\left[-\varepsilon - \frac{1}{\sigma}\right]\right)$. $\square$

We therefore work under Assumption 3.4 for the rest of this section. The remaining analysis reasons about specific spheres $S_{R,1}$.

**Definition 3.16.** *For $R > 0$, define $U_R$ to be the fraction of $S_{R,1}$ contained in $V$.*

We start by identifying when $U_R = 0$.

**Lemma 3.17.** *If $0 < R < \frac{1+\varepsilon\sigma}{2}$, then $U_R = 0$.*

*Proof.* Shorthand $\tau = \varepsilon\sigma$ for neatness. By Corollary 3.7, $F_{r,h(r)} = 1$ for $r \leq \frac{1-\tau}{2}$. Let $r_1 = \frac{1-\tau}{2}$ denote the largest radius such that $F_{r,h(r)} = 1$. This shows that the point with the largest $e_1$-coordinate in $V \cap S_{r,0}$ has $e_1$-coordinate $r$ for $0 < r \leq r_1$. For $r > r_1$, the point with the largest $e_1$-coordinate in $V \cap S_{r,0}$ has $e_1$-coordinate $-r + h(r) = -r\tau + \frac{1-\tau^2}{2}$ which is monotonically decreasing as $r$ increases. Overall, the point with the largest $e_1$-coordinate in $V \cap S_{r,0}$ is increasing for $0 < r \leq r_1$ and decreasing for $r > r_1$, so $r_1$ is the maximum $e_1$-coordinate of this point over all radii.

It follows that $0 < R < 1 - r_1$ implies $S_{R,1} \cap S_{r,0} = \emptyset$ for all $r > 0$, and since $V \subset \cup_{r \in \mathbb{R}^+} S_{r,0}$, we get $U_R = 0$. $\square$

It remains to handle the large $R$ case. We will show that, as was the case in the upper bound analysis, each $S_{R,1} \cap V$ is a spherical cap on $S_{R,1}$. The first result proves that $S_{R,1} \cap V$ has this form. This result is not technically necessary for the rest of the argument, but it explains why we attempt to solve for $S_{R,1} \cap V$ as a cap later.

**Lemma 3.18.** *For $R \geq \frac{1+\varepsilon\sigma}{2}$, $S_{R,1} \cap V$ is a spherical cap $\hat{S}_{R,1,H(R)}$.*

*Proof.* Shorthand $\tau = \varepsilon\sigma$. Recall from Lemma 3.2 that $V$ is the convex hull of the $-e_1$ facing component of a hyperboloid with foci 0 and $e_1$ and with constant difference $\tau$. To see why $S_{R,1} \cap V$ is a spherical cap, observe that both the hyperboloid-bounded region $V$ and $S_{R,1}$ are symmetric around $e_1$, so their intersection must be symmetric around $e_1$ as well. Let $P_{v_1, v_2}$ denote projection onto $\mathrm{span}(v_1, v_2)$. Then $P_{e_1, e_2}(V) \cap P_{e_1, e_2}(S_{R,1})$ is a 1-dimensional spherical cap of some height $H$ in $P_{e_1, e_2}(S_{R,1})$. Since $V \cap S_{R,1}$ is symmetric around $e_1$, the previous sentence also holds if we

replace $P_{e_1,e_2}$ with $P_{e_1,v}$ for any $v$ orthogonal to $e_1$. Thus $V \cap S_{R,1} = \cup_{v \perp e_1} P_{e_1,v}(V) \cap P_{e_1,v}(S_{R,1}) = \hat{S}_{R,1,H}$. $\square$

It remains to identify the $H$ referenced in Lemma 3.18. To do so, we start with an arbitrary $y \in S_{R,1}$ and solve for an $e_1$ coordinate $X$ such that $y_1 \leq X$ if and only if $y_1 \leq -\|y\|_2 + h(\|y\|_2)$ (Lemma 3.6). The bulk of the proof beyond this idea is algebraic manipulation, so it appears in Appendix A.2.

**Lemma 3.19.** *Define* $X = \frac{1+(\varepsilon\sigma)^2 - 2\varepsilon\sigma R}{2}$ *and* $H(R) = R - 1 + X$. *Then cap* $\hat{S}_{R,1,H(R)} = S_{R,1} \cap V$.

**Lemma 3.20.** *If* $R \geq \frac{1+\varepsilon\sigma}{2}$*, then* $U_R$ *is monotone increasing in* $R$.

*Proof.* Shorthand $\tau = \varepsilon\sigma$. Lemma 3.17 established that $U_R = 0$ for $R < \frac{1+\tau}{2}$, so it remains to show that $U_R$ is increasing in $R$ for $R \geq \frac{1+\tau}{2}$. The proof of Lemma 3.19 established that $H(R) \in [0, R]$, so we apply Lemma 3.9 to get that $\hat{S}_{R,1,H(R)}$ occupies a fraction of $S_{R,1}$ given by $\frac{1}{2} I_{(2RH(R) - H(R)^2)/R^2} \left(\frac{d-1}{2}, \frac{1}{2}\right)$. By the same logic used in the proof of Lemma 3.10, we can show that $U_R$ is monotone increasing by verifying that the expression in the $I$ subscript is nonnegative and increasing in $R$.

The first condition follows from the aforementioned result $H(R) \leq R$. For the second condition,

$$\frac{2R[R-1+X] - [R-1+X]^2}{R^2}$$
$$= \frac{R^2 - 1 + 2X - X^2}{R^2}$$
$$= 1 - \left(\frac{X-1}{R}\right)^2$$
$$= 1 - \left(\frac{\tau^2 - 2\tau R - 1}{2R}\right)^2.$$

We wanted to prove that this expression is increasing in $R$, so we show that the second term is decreasing in $R$. Taking its derivative with respect to $R$ yields

$$2\left(\frac{\tau^2 - 2\tau R - 1}{2R}\right) \cdot \frac{2R(-2\tau) - 2(\tau^2 - 2\tau R - 1)}{4R^2}$$
$$= \left(\frac{\tau^2 - 2\tau R - 1}{R}\right) \cdot \frac{1 - \tau^2}{2R^2}.$$

Because $R > 0$ and $0 < \tau < 1$, the numerator of the first term is negative, and the remaining terms are positive, so the entire quantity is negative. $\square$

Since we want to lower bound the mass on these $U_R$, we use Lemma 3.12 and employ a left Riemann sum in which the $j$th approximating rectangle has base length as the $j$th

interval on the grid $\left[\frac{\gamma(d, R_1/\sigma)}{\Gamma(d)}, ..., \frac{\gamma(d, R_{n_R}/\sigma)}{\Gamma(d)}, \infty\right]$ and has height $F_{R_j, H(R_j)}$. The proof of the following result uses similar logic as the proof of Lemma 3.13.

**Lemma 3.21.** *Suppose Assumption 3.4 holds. Let* $R_1 < \ldots < R_{n_R}$ *where* $R_1 = \frac{1+\tau}{2}$*. Then for* $y \sim M(1)$*,*

$$\mathbb{P}[y \in V] \geq \sum_{j=1}^{n_R - 1} \frac{\gamma(d, R_{j+1}/\sigma) - \gamma(d, R_j/\sigma)}{\Gamma(d)} F_{R_j, H(R_j)}$$
$$+ \frac{\Gamma(d, R_{n_R}/\sigma)}{\Gamma(d)} F_{R_{n_R}, H(R_{n_R})}$$

*where we reused the definition of $F$ from Definition 3.8.*

Algorithm 2 provides overall lower bound pseudocode.

---
**Algorithm 2** Term2LowerBound
---
1: **Input:** Dimension $d$; scale parameter $\sigma$; privacy parameter $\varepsilon$; largest radius $R^*$; number of radii $n_R$
2: **if** $\sigma \geq 1/\varepsilon$ **then**
3:     Return 0 (Corollary 3.14)
4: **end if**
5: **if** $d = 1$ **then**
6:     Return $\frac{1}{2} \exp\left(\frac{1}{2}\left[-\varepsilon - \frac{1}{\sigma}\right]\right)$ (Lemma 3.15)
7: **end if**
8: Define $R_1 \leftarrow \frac{1+\tau}{2}$, $R_{n_R} \leftarrow R^*$, and $R_2, \ldots, R_{n_R-1}$ regularly spaced between $R_1$ and $R_{n_R}$
9: Return lower bound from Lemma 3.21
---

### 3.1.3. OVERALL ALGORITHM

Algorithm 1 upper bounds the first term in the inequality in Lemma 2.3 and Algorithm 2 lower bounds the second term. This upper bounds the LHS of the inequality, so if it is at most $\delta$, then the mechanism is $(\varepsilon, \delta)$-DP.

The last step is choosing $n_r$, $r^*$, $n_R$, and $R^*$. Our experiments suggests that setting $n_r = n_R = 1000$ yields a reasonably tight approximation for $d \leq 100$ (see Section 4.1); larger values should only be tighter, at the cost of speed. We choose $r^*$ using Lemma 3.12 so $\mathbb{P}_{y \sim M(0)}[\|y\|_2 > r^*] = \frac{\delta}{100}$; in the context of Lemma 3.13, we use $F_{r_{n_r}, h(r_{n_r})}$ to upper bound the cap fraction for all $S_{r,0}$ with $r \geq r_{n_r}$, so we choose $r^*$ to make the effect of this approximation negligible. By the same logic, we use $R^* = r^*$.

Algorithm 3 collects the entire process into pseudocode.

### 3.2. Parallel Sampler

The sampler described here is a simple consequence of Lemma 2.4 and well-known statistical facts, and similar samplers have appeared for related mechanisms (Yu et al., 2014; Steinke & Ullman, 2016). We collect the relevant information here, with proofs in Appendix A.3 for completeness.

**Algorithm 3** CheckApproximateDP

1: **Input:** Dimension $d$; scale parameter $\sigma$; privacy parameters $\varepsilon, \delta$; numbers of radii $n_r, n_R$
2: Compute $r^*$ such that $\mathbb{P}_{y \sim M(0)}[\|y\|_2 > r^*] = \frac{\delta}{100}$ (Lemma 3.12)
3: $T_1 \leftarrow \text{Term1UpperBound}(d, \sigma, \varepsilon, r^*, n_r)$
4: $T_2 \leftarrow \text{Term2LowerBound}(d, \sigma, \varepsilon, r^*, n_R)$
5: Return $(T_1 - e^\varepsilon T_2 \le \delta)$

---

By Lemma 2.4, we sample $rz$ where $r \sim \text{Gamma}(d+1, \sigma)$ and $z \sim_U B_2^d$. In a parallel setting, suppose we have a worker for each of $d$ coordinates and a central manager. We first sample $r$.

**Lemma 3.22.** *Let $U_1, \ldots, U_{d+1} \sim_{iid} U(0,1)$ be uniform random samples. Then $-\sigma \sum_{i=1}^{d+1} \log(U_i) \sim \text{Gamma}(d+1, \sigma)$.*

By Lemma 3.22, each worker $i$ can sample $\log(U_i)$, and the manager can add the combined sum to their own sample and scale the result by $-\sigma$ to obtain $r$. It remains to sample $z$.

**Lemma 3.23.** *Let $X_1, \ldots, X_d \sim_{iid} N(0,1)$, and let $Y \sim U(0,1)$ be a uniform sample from $[0,1]$. Then $Y^{1/d} \cdot \frac{(X_1, \ldots, X_d)}{\sqrt{\sum_{i=1}^{d} X_i^2}}$ is a uniform sample from $B_2^d$.*

To apply Lemma 3.23, each worker samples a standard Gaussian and reports its square in the same combine used to compute $r$ in Lemma 3.22. The manager samples $Y$ and publishes it along with $r$ and the sum of squares. At this point, each worker can compute their coordinate of $rz$.

## 4. Experiments

This section discusses experiments evaluating the tightness of our privacy analysis (Section 4.1) as well as the $\ell_2$ mechanism's error (Section 4.2) and speed (Section 4.3). All experiments use the $\ell_2$ mechanism with $n_r = n_R = 1000$. Experiment code may be found on Github (Google, 2025).

### 4.1. Privacy

Our first experiments attempt to measure the tightness of our privacy analysis. To do so, we compare two methods for estimating $\mathbb{P}[\ell_{M,X,X'} \ge \varepsilon] - e^\varepsilon \mathbb{P}[\ell_{M,X',X} \le -\varepsilon]$, the quantity that must be upper bounded by $\delta$ for $M$ to satisfy $(\varepsilon, \delta)$-DP (Lemma 2.3).

We fix $\varepsilon = 1$, $\delta = 0.01$, and vary $d = 1, 2, \ldots, 100$. At each $d$, the first method empirically estimates the smallest $\sigma$ such that

$$\mathbb{P}[\ell_{M,X,X'} \ge 1] - e \cdot \mathbb{P}[\ell_{M,X',X} \le -1] \le \delta, \quad (3)$$

where $M$ is the $\ell_2$ mechanism with parameter $\sigma$, and $M(X)$ is centered at 0 while $M(X')$ is centered at 1. By the

analysis of Section 3, Equation (3) is equivalent to

$$\mathbb{P}_{y \sim M(0)} \left[ \frac{1}{\sigma}(\|e_1 - y\|_2 - \|y\|_2) \ge \varepsilon \right]$$
$$- e \cdot \mathbb{P}_{y \sim M(1)} \left[ \frac{1}{\sigma}(\|e_1 - y\|_2 - \|y\|_2) \ge \varepsilon \right] \le \delta.$$

We can estimate the value of this expression by drawing $n$ samples from $M(0)$ and $n$ samples from $M(1)$, counting the fraction $c_1$ satisfying the first inequality and the fraction $c_2$ satisfying the second inequality, and then computing $c_1 - e \cdot c_2$. Binary searching over $\sigma$ to find the smallest value where this computation is upper bounded by $\delta$ leads to an empirical estimate of the minimum $\sigma$ yielding a $(1, 0.01)$-DP $\ell_2$ mechanism.

The second method computes its $\sigma$ using the analysis of Section 3. As shown in Figure 4, our algorithm closely tracks the empirical values. This provides empirical evidence that the resulting privacy analysis is both sound and (with the chosen $n_r = n_R = 1000$ and range for $d$) tight.

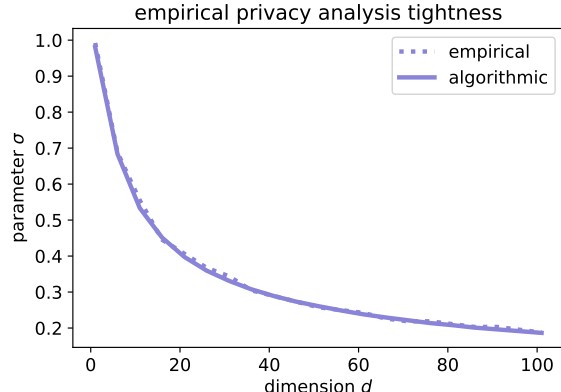

*Figure 4.* A comparison of empirical (dotted) and algorithmic (solid) estimates of privacy loss. At each $d$, the empirical method uses $n = 1000/\delta = 10^5$ samples.

### 4.2. Error

We first derive some basic results about the mean squared $\ell_2$ error of the Laplace, $\ell_2$, and Gaussian mechanisms used in our experiments. The Laplace and $\ell_2$ results are corollaries of the following lemma. The lemma is an extension of a previous result about the mean squared $\ell_2$ norm of a sample from an $\ell_p$ ball (Joseph et al., 2025) and is proved in Appendix A.4.

**Lemma 4.1.** *The mean squared $\ell_2$ error of the $d$-dimensional $\ell_p$ mechanism with parameter $\sigma$ is*

$$(d\sigma)^2 (d+1) \left( \frac{\Gamma(\frac{d}{p})\Gamma(\frac{3}{p})}{\Gamma(\frac{1}{p})\Gamma(\frac{d+2}{p})} \right).$$

Since a $d$-dimensional statistic with $\ell_2$ sensitivity 1 has $\ell_1$ sensitivity $\sqrt{d}$, substituting $p = 1$ and parameter $\sigma\sqrt{d}$ into Lemma 4.1 yields the following result for the Laplace mechanism.

**Corollary 4.2.** *The Laplace mechanism with parameter $\sigma\sqrt{d}$ has mean squared $\ell_2$ error $2d^2\sigma^2$.*

The $\ell_2$ mechanism uses $p = 2$ and parameter $\sigma$.

**Corollary 4.3.** *The $\ell_2$ mechanism with parameter $\sigma$ has mean squared $\ell_2$ error $d(d + 1)\sigma^2$.*

A similar result for the Gaussian mechanism is easy to prove directly (see Appendix A.4 for proof).

**Lemma 4.4.** *The Gaussian mechanism with parameter $\sigma$ has mean squared $\ell_2$ error $d\sigma^2$.*

For a range of $d$, we solve for the smallest possible $\sigma$ for each mechanism to achieve $(\varepsilon, \delta)$-DP and plot the mean squared $\ell_2$ error according to the preceding results. The Laplace mechanism uses $\sigma = \sqrt{d}/(\varepsilon + \delta)^2$, the Gaussian mechanism binary searches over $\sigma$ as described by Balle & Wang (2018), and the $\ell_2$ mechanism binary searches over $\sigma$ using the algorithms from Section 3. Throughout, binary searches use tolerance 0.001 and we use $(1, 10^{-5})$-DP.

This produces the left plot in Figure 1 in the introduction. The Laplace mechanism obtains lower error than the analytic Gaussian mechanism for small $d$, the analytic Gaussian mechanism obtains lower error than the Laplace mechanism for larger $d$, and the $\ell_2$ mechanism dominates both. The gap between the $\ell_2$ mechanism and the better of the Laplace mechanism and analytic Gaussian mechanism is 0 at $d = 1$ (when the Laplace and $\ell_2$ mechanism are identical) and peaks at 50% at $d = 7$ before gradually shrinking, to 5% at $d = 100$ and $< 1\%$ at $d = 500$ (not pictured).

Analogous plots for a high-privacy regime of $(0.1, 10^{-7})$-DP and a low-privacy regime of $(10, 10^{-3})$-DP are essentially the same.

## 4.3. Speed

The last set of experiments evaluates the speed of the $\ell_2$ mechanism, as executed on a typical personal computer. This runtime is split into two operations: the time to compute $\sigma$ and the time to sample the mechanism.

The largest gap appears in the time to compute $\sigma$ (Figure 5). The Laplace computation is $\approx 100\text{x}$ faster than the Gaussian computation, which is $\approx 100\text{x}$ faster than the $\ell_2$ computation. This may be expected, as the Laplace computation is a single arithmetic expression, the Gaussian computation is a binary search over the standard normal CDF, and the

---

$\ell_2$ computation is a binary search where each evaluation iterates over $n_r + n_R = 2000$ radii. Nonetheless, we note that the $\ell_2$ computation still runs in $\approx 0.1$ seconds, this time does not increase with $d$, and the calculation only needs to be performed once for each setting of $(\varepsilon, \delta, d)$.

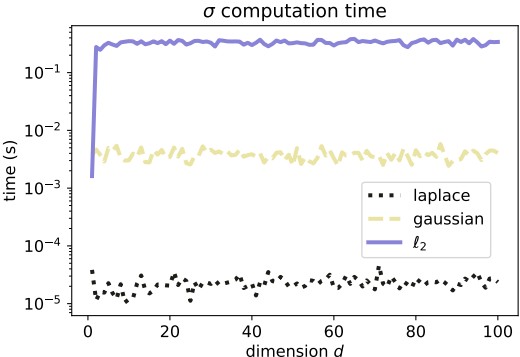

*Figure 5.* A plot of time in seconds to compute the minimum $\sigma$ to achieve $(1, 10^{-5})$-DP. The $\ell_2$ mechanism line jumps after $d = 1$ because that case uses Lemma 3.3 instead of approximating the spherical cap region.

The time to draw 1000 mechanism samples is less varied (Figure 6). The $\ell_2$ mechanism is again slowest, but it is within a factor of two of the other mechanisms, and no mechanism takes more than $\approx 0.01$ seconds.

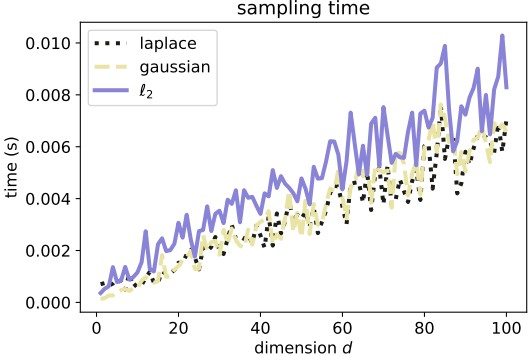

*Figure 6.* This plot uses the same setup as Figure 5 but records sampling time.

---

[2]Note that the smallest possible $\sigma$ for which the Laplace mechanism is $(\varepsilon, \delta)$-DP is provably negligibly smaller than the one used here. See Appendix A.4 for details.

# Impact Statement

This paper presents work whose goal is to advance the field of Machine Learning. There are many potential societal consequences of our work, none which we feel must be specifically highlighted here.

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

# A. Appendix

## A.1. Omitted Proofs From Upper Bound

**Claim 3.6.1.** $\|y - e_1\|_2 - \|y\|_2$ *decreases as $\theta$ decreases.*

*Proof.* By the law of cosines,

$$\|y - e_1\|_2 = \sqrt{\|y\|_2^2 + \|e_1\|_2^2 - 2\|y\|_2\|e_1\|_2 \cos(\theta)}$$
$$= \sqrt{r^2 + 1 - 2r\cos(\theta)},$$

so as $y$ moves clockwise through $S_{r,0} \cap H$ from $-e_1$ to $e_1$, $\theta$ decreases from $\pi$ to $0$, $\cos(\theta)$ grows from $-1$ to $1$, and $\|y - e_1\|_2$ shrinks from $r + 1$ to $|r - 1|$. Since $\|y\|_2 = r$ remains constant, $\|y - e_1\|_2 - \|y\|_2$ decreases as $\theta$ decreases. The same conclusion holds if we choose $y$ in the lower half plane and consider the analogous counterclockwise angle. $\square$

**Claim 3.6.2.** $\frac{1}{\sigma}(\|p - e_1\|_2 - \|p\|_2) = \varepsilon$ *is equivalent to* $h(r) = r(1 - \varepsilon\sigma) + \frac{1 - \varepsilon^2\sigma^2}{2}$.

*Proof.*

$$\|p - e_1\|_2 = \sqrt{(r + 1 - h(r))^2 + 2h(r)r - h(r)^2}$$
$$= \sqrt{(r + 1)^2 - 2h(r)}$$

and

$$\varepsilon = \frac{1}{\sigma}(\|p - e_1\|_2 - \|p\|_2)$$
$$\varepsilon = \frac{1}{\sigma}(\sqrt{(r + 1)^2 - 2h(r)} - r)$$
$$(\sigma\varepsilon + r)^2 = (r + 1)^2 - 2h(r)$$
$$2h(r) = 2r + 1 - (\sigma\varepsilon)^2 - 2r\sigma\varepsilon$$
$$h(r) = r(1 - \varepsilon\sigma) + \frac{1 - \varepsilon^2\sigma^2}{2}.$$

$\square$

**Claim 3.6.3.** $r(1 - \varepsilon\sigma) + \frac{1 - \varepsilon^2\sigma^2}{2} \in [0, 2r]$ *if and only if* $r \geq \frac{1 - \varepsilon\sigma}{2}$.

*Proof.* The upper constraint of $r(1 - \varepsilon\sigma) + \frac{1 - \varepsilon^2\sigma^2}{2} \leq 2r$ is equivalent to $r \geq \frac{1 - \varepsilon\sigma}{2}$ as follows

$$r(1 - \varepsilon\sigma) + \frac{1 - \varepsilon^2\sigma^2}{2} \leq 2r$$
$$\frac{1 - \varepsilon^2\sigma^2}{2} \leq r(1 + \varepsilon\sigma)$$
$$\frac{1 - \varepsilon\sigma}{2} \leq r.$$

The lower constraint of $r(1 - \varepsilon\sigma) + \frac{1 - \varepsilon^2\sigma^2}{2} \geq 0$ is satisfied for any $r$ since $1 - \varepsilon\sigma \geq 0$ implies

$$r(1 - \varepsilon\sigma) + \frac{1 - \varepsilon^2\sigma^2}{2} = (1 - \varepsilon\sigma)\left(r + \frac{1 + \varepsilon\sigma}{2}\right) \geq 0.$$

$\square$

**Lemma 3.12.** *For $r > 0$, $\mathbb{P}_{y \sim M(0)}[\|y\|_2 \leq r] = \frac{\gamma(d, r/\sigma)}{\Gamma(d)}$.*

*Proof.* The density for $Y$ is $f(y) \propto \exp\left(-\|y\|_2/\sigma\right)$, so we compute the distribution's normalization factor $Z$. We use two facts. First, a $(d-1)$-sphere, i.e., a sphere in $\mathbb{R}^d$, with radius $s$ has surface area $\frac{2\pi^{d/2}}{\Gamma(d/2)} \cdot s^{d-1}$. Second, by $u$-substitution with $u = s/\sigma$,

$$\int_0^\infty e^{-s/\sigma} s^{d-1} ds = \int_0^\infty e^{-u} \cdot u^{d-1}\sigma^d du = \Gamma(d)\sigma^d$$

since $\Gamma(z) = \int_0^\infty e^{-t}t^{z-1}dt$. We compute the integral $Z$ using hyperspherical coordinates $s, \theta_1, ..., \theta_{d-1}$ where $s \geq 0$, $\theta_1 \in [0, 2\pi]$, and $\theta_j \in [0, \pi]$ for $2 \leq j \leq d-1$. Let

$$V(\theta_1, ..., \theta_{d-1}) = \sin^{d-2}(\theta_1) \sin^{d-3}(\theta_2)... \sin(\theta_{d-1})$$

be the angle dependent terms of the hyperspherical volume element. Then

$$\begin{aligned}
Z &= \int_0^\infty \int_0^{2\pi} \int_0^\pi ... \int_0^\pi e^{-s/\sigma} s^{d-1} V(\theta_1, ..., \theta_{d-1})\partial_{d-1} ... \partial_1 \partial s \\
&= \frac{2\pi^{d/2}}{\Gamma(d/2)} \int_0^\infty e^{-s/\sigma} s^{d-1} \partial s \\
&= \frac{2\pi^{d/2}\sigma^d}{\Gamma(d/2)} \cdot \Gamma(d).
\end{aligned}$$

This gives

$$\begin{aligned}
\mathbb{P}_Y\left[\|y\| \leq r\right] &= \frac{1}{Z} \cdot \frac{2\pi^{d/2}}{\Gamma(d/2)} \int_0^r e^{-s/\sigma} s^{d-1} ds \\
&= \frac{1}{Z} \cdot \frac{2\pi^{d/2}\sigma^d}{\Gamma(d/2)} \cdot \gamma(d, r/\sigma) \\
&= \frac{\gamma(d, r/\sigma)}{\Gamma(d)}.
\end{aligned}$$

$\square$

**Lemma 3.10.** *$F_{r,h(r)}$ is monotone decreasing in $r$.*

*Proof.* Shorthand $\tau = \varepsilon\sigma$. By Corollary 3.7, $F_{r,h(r)} = 1$ for $r \leq \frac{1-\tau}{2}$. Suppose $r > \frac{1-\tau}{2}$. Then by Lemma 3.6, $h(r) = r(1-\tau) + \frac{1-\tau^2}{2}$.

Case 1: $r < \frac{1-\tau^2}{2\tau}$. Then $h(r) > r$, and $F_{r,h(r)} = 1 - F_{r,2r-h(r)}$. Since we want to prove that $F_{r,h(r)}$ decreases with $r$, it suffices to show that $F_{r,2r-h(r)}$ increases with $r$. By Lemma 3.9,

$$F_{r,2r-h(r)} = \frac{1}{2} I_{(2r[2r-h(r)]-[2r-h(r)]^2)/r^2}\left(\frac{d-1}{2}, \frac{1}{2}\right).$$

We expand the subscript for $I$

$$\begin{aligned}
\frac{2r(2r-h(r)) - (2r-h(r))^2}{r^2} &= \frac{4r^2 - 2rh(r) - (4r^2 - 4rh(r) + h(r)^2)}{r^2} \\
&= \frac{2rh(r) - h(r)^2}{r^2}.
\end{aligned} \tag{4}$$

Since $h(r) \in [0, 2r]$ (Lemma 3.6), $2rh(r) - h(r)^2 \geq 0$. Because $I_x(a, b)$ increases with $x$ for $x \geq 0$, it is enough to show that $[2rh(r) - h(r)^2]/r^2$ increases with $r$. Expanding yields

$$\frac{2rh(r) - h(r)^2}{r^2} = \frac{2r(1-\tau) + (1-\tau^2)}{r} - \frac{r^2(1-\tau)^2 + r(1-\tau)(1-\tau^2) + \frac{(1-\tau^2)^2}{4}}{r^2}.$$

We drop terms that don't depend on $r$ to get

$$\frac{1-\tau^2}{r} - \frac{(1-\tau)(1-\tau^2)}{r} - \frac{(1-\tau^2)^2}{4r^2} = (1-\tau^2)\left[\frac{\tau}{r} - \frac{(1-\tau^2)}{4r^2}\right].$$

Differentiating the second term with respect to $r$ gives $\frac{1-2\tau r-\tau^2}{2r^3}$, and this is positive exactly when $r < \frac{1-\tau^2}{2\tau}$.

Case 2: $r \geq \frac{1-\tau^2}{2\tau}$. Then $h(r) \leq r$, and

$$F_{r,h(r)} = \frac{1}{2}I_{(2rh(r)-h(r)^2)/r^2}\left(\frac{d-1}{2},\frac{1}{2}\right).$$

By similar logic, it suffices to show that $(2rh(r) - h(r)^2)/r^2$ is nonincreasing in $r$ for $r \geq \frac{1-\tau^2}{2\tau}$. This follows from the analysis of the previous case. $\square$

## A.2. Omitted Proofs From Lower Bound

**Lemma 3.19.** *Define $X = \frac{1+(\varepsilon\sigma)^2-2\varepsilon\sigma R}{2}$ and $H(R) = R - 1 + X$. Then cap $\hat{S}_{R,1,H(R)} = S_{R,1} \cap V$.*

*Proof.* Shorthand $\tau = \varepsilon\sigma$ for neatness. To verify the claim, we start with an arbitrary $y \in S_{R,1}$ and attempt to determine a cutoff $X \in \mathbb{R}$ such that $y \in V$ iff $y_1 \leq X$. For any two points $y, y' \in S_{R,1}$ such that $y_1 = y'_1$, it is true that $y \in V$ iff $y' \in V$ since $V$ is spherically symmetric around $e_1$. If $y' = y_1 e_1 + v$ for some $v$ orthogonal to $e_1$, then by $S_{R,1}$'s spherical symmetry around $e_1$, the point $y = y_1 e_1 + |v|e_2$ is also in $S_{R,1}$. Therefore, our goal is to find the minimum cutoff $X$ for the point $y = (y_1, y_2, 0, ..., 0)$ such that $y \in V$ iff $y_1 \leq X$.

We know $y \in S_{r',0}$ for some $r' > 0$. Since $y_1^2 + y_2^2 = r'^2$, and $y \in S_{R,1}$ implies $(y_1 - 1)^2 + y_2^2 = R^2$, then combining these yields $r' = \sqrt{R^2 + 2y_1 - 1}$. Thus we have $y \in V$ if and only if $y_1 \leq -r' + h(r')$. By Lemma 3.6, $-r' + h(r') = \min(-\tau r' + \frac{1-\tau^2}{2}, 2r')$. We have $-\tau r' + \frac{1-\tau^2}{2} = -\tau\sqrt{R^2 + 2y_1 - 1} + \frac{1-\tau^2}{2}$, so we solve for the largest $X$ where $X \leq \min(-\tau\sqrt{R^2 + 2X - 1} + \frac{1-\tau^2}{2}, 2\sqrt{R^2 + 2X - 1})$.

Solving for $X$ under the first constraint yields

$$\left(X - \frac{1-\tau^2}{2}\right)^2 \geq \tau^2(R^2 + 2X - 1)$$

$$X^2 - X(1+\tau^2) + \frac{\tau^4 - 2\tau^2 + 1 - 4\tau^2 R^2 + 4\tau^2}{4} \geq 0$$

$$X^2 - X(1+\tau^2) + \frac{\tau^4 + 2\tau^2 + 1 - 4\tau^2 R^2}{4} \geq 0. \tag{5}$$

The roots of the LHS are given by

$$X = \frac{1+\tau^2 \pm \sqrt{(1+\tau^2)^2 - ([1+\tau^2]^2 - 4\tau^2 R^2)}}{2} = \frac{1+\tau^2 \pm 2\tau R}{2}.$$

Let $x_1 = \frac{1+\tau^2-2\tau R}{2}$ and $x_2 = \frac{1+\tau^2+2\tau R}{2}$. As the LHS of Equation (5) is a convex parabola, the inequality is satisfied on the intervals $(-\infty, x_1] \cup [x_2, \infty)$. But the first constraint on $X$ also implies the weaker inequality $X < \frac{1-\tau^2}{2}$ so $X \notin [x_2, \infty)$. Then $x_1$ is the largest value that satisfies the first constraint.

For any $X \in (x_1, x_2)$, we have $X > -\tau\sqrt{R^2 + 2X - 1} + \frac{1-\tau^2}{2} \geq \min(-\tau\sqrt{R^2 + 2X - 1} + \frac{1-\tau^2}{2}, 2\sqrt{R^2 + 2X - 1})$. So if we can show that $x_1 \leq 2\sqrt{R^2 + 2x_1 - 1}$, then $x_1$ will indeed be the desired cutoff. We actually prove a stronger inequality

$$x_1 \leq \sqrt{R^2 + 2x_1 - 1}$$
$$\frac{1+\tau^2 - 2\tau R}{2} \leq R - \tau$$
$$(1+\tau)^2 \leq 2R(1+\tau)$$
$$\frac{1+\tau}{2} \leq R$$

which follows from our starting assumption on $R$. So $X = \frac{1+\tau^2-2\tau R}{2}$ is the desired cutoff. This leads to a cap on $S_{R,1}$ of height

$$H(R) = X - (1-R) = \frac{1+\tau^2 - 2\tau R}{2} - 1 + R = R(1-\tau) - \frac{1-\tau^2}{2}.$$

The last step is verifying that this is a valid height lying in $[0, 2R]$. The lower bound follows from $R(1-\tau) \geq \frac{1-\tau^2}{2}$ rearranging into the starting assumption $R \geq \frac{1+\tau}{2}$. We prove a stronger upper bound of $R$ by rearranging

$$R(1-\tau) - \frac{1-\tau^2}{2} \leq R$$

$$-\frac{1-\tau^2}{2} \leq \tau R$$

which uses $0 < \tau < 1$ and $R > 0$. $\qquad\square$

### A.3. Omitted Proofs From Sampler

**Lemma 3.22.** *Let $U_1, \ldots, U_{d+1} \sim_{iid} U(0,1)$ be uniform random samples. Then $-\sigma \sum_{i=1}^{d+1} \log(U_i) \sim \mathsf{Gamma}\,(d+1, \sigma)$.*

*Proof.* We first show that $-\log(U(0,1)) \sim \mathsf{Expo}\,(1)$, an exponential random variable. Let $f$ be the CDF of $\mathsf{Expo}\,(1)$. Then $\mathbb{P}[f^{-1}(U) \leq t] = \mathbb{P}[U \leq f(t)] = f(t)$ so $f^{-1}(U) \sim \mathsf{Expo}\,(1)$. Since $f^{-1}(t) = -\log(1-t)$ for $0 \leq t \leq 1$, and $U \sim (1-U)$, we get $-\log(U) \sim \mathsf{Expo}\,(1)$.

Note that $\mathsf{Expo}\,(1)$ corresponds to a random variable that measures the time required for the first arrival from a Poisson process with rate 1. Moreover, $\mathsf{Gamma}\,(d+1, \sigma) \sim \sigma \mathsf{Gamma}\,(d+1, 1)$ and $\mathsf{Gamma}\,(d+1, 1)$ corresponds to a random variable that measures the time of the $(d+1)$th arrival of a Poisson process with rate 1. The random variable of the $(d+1)$th arrival is equal to the sum of the random variables of interarrival times for the first $(d+1)$ arrivals. Since a Poisson process has stationary increments, each of these interarrival times are i.i.d. as $\mathsf{Expo}\,(1)$. It follows that $\mathsf{Gamma}\,(d+1, 1) \sim \sum_{i=1}^{d+1} E_i \sim -\sum_{i=1}^{d+1} \log(U_i)$ where $E_i \sim \mathsf{Expo}\,(1)$. $\qquad\square$

**Lemma 3.23.** *Let $X_1, \ldots, X_d \sim_{iid} N(0,1)$, and let $Y \sim U(0,1)$ be a uniform sample from $[0,1]$. Then $Y^{1/d} \cdot \frac{(X_1,\ldots,X_d)}{\sqrt{\sum_{i=1}^d X_i^2}}$ is a uniform sample from $B_2^d$.*

*Proof.* The term $\frac{(X_1,\ldots,X_d)}{\sqrt{\sum_{i=1}^d X_i^2}}$ is a normalized draw from a $d$-dimensional multivariate Gaussian with an identity covariance matrix. As this distribution is spherically symmetric, normalizing the draw to have unit length produces a uniform draw from the unit sphere. Define the function $f$ to be the CDF of the random variable of the $\ell_2$ norm of a uniform sample from $B_2^d$. Then $f(r) = r^d$. We show that $f$ is also the CDF of $Y^{1/d}$. We have $\mathbb{P}[Y^{1/d} \leq r] = \mathbb{P}[U(0,1)^{1/d} \leq r] = \mathbb{P}[U(0,1) \leq r^d] = r^d$, and the lemma follows. $\qquad\square$

### A.4. Omitted Proofs From Experiments

The following result about the expected squared $\ell_2$ norm of $\ell_p$ balls will be useful.

**Lemma A.1** ((Joseph et al., 2025)). *Let $\mathbb{E}_2^2(X)$ denote the expected squared $\ell_2$ norm of a uniform sample from $X$, and let $rB_p^d$ denote the $d$-dimensional $\ell_p$ ball of radius $r$. Then $\mathbb{E}_2^2(rB_p^d) = r^2 \cdot \frac{d}{3}\left(\frac{3d}{d+2}\right)\left(\frac{\Gamma(\frac{d}{p})\Gamma(\frac{3}{p})}{\Gamma(\frac{1}{p})\Gamma(\frac{d+2}{p})}\right).$*

**Lemma 4.1.** *The mean squared $\ell_2$ error of the $d$-dimensional $\ell_p$ mechanism with parameter $\sigma$ is*

$$(d\sigma)^2(d+1)\left(\frac{\Gamma(\frac{d}{p})\Gamma(\frac{3}{p})}{\Gamma(\frac{1}{p})\Gamma(\frac{d+2}{p})}\right).$$

*Proof.* Consider the mechanism releasing a noisy version of $T(X) = 0$. Call this mechanism $M_\sigma^p$. Recall from Lemma 2.4 that we can sample it by sampling $r \sim \mathsf{Gamma}\,(d+1, \sigma)$, sampling $z \sim B_p^d$, and outputting $rz$. The distribution $\mathsf{Gamma}\,(d+1, \sigma)$ has density

$$f(x) = \frac{x^d e^{-x/\sigma}}{\Gamma(d+1)\sigma^{d+1}}. \tag{6}$$

so

$$\begin{aligned}
\mathbb{E}_{y \sim M_\sigma^p}\left[\|y\|_2^2\right] &= \int_0^\infty f(r)\mathbb{E}_2^2(rB_p^d)dr \\
&= \int_0^\infty \frac{r^d e^{-r/\sigma}}{\Gamma(d+1)\sigma^{d+1}}r^2 \cdot \frac{d}{3}\left(\frac{3d}{d+2}\right)\left(\frac{\Gamma(\frac{d}{p})\Gamma(\frac{3}{p})}{\Gamma(\frac{1}{p})\Gamma(\frac{d+2}{p})}\right)dr \\
&= \frac{d}{3\Gamma(d+1)\sigma^{d+1}}\left(\frac{3d}{d+2}\right)\left(\frac{\Gamma(\frac{d}{p})\Gamma(\frac{3}{p})}{\Gamma(\frac{1}{p})\Gamma(\frac{d+2}{p})}\right)\int_0^\infty r^{d+2}e^{-r/\sigma}dr \\
&= \frac{d}{3\Gamma(d+1)\sigma^{d+1}}\left(\frac{3d}{d+2}\right)\left(\frac{\Gamma(\frac{d}{p})\Gamma(\frac{3}{p})}{\Gamma(\frac{1}{p})\Gamma(\frac{d+2}{p})}\right)\cdot \Gamma(d+3)\sigma^{d+3} \\
&= (d\sigma)^2(d+1)\left(\frac{\Gamma(\frac{d}{p})\Gamma(\frac{3}{p})}{\Gamma(\frac{1}{p})\Gamma(\frac{d+2}{p})}\right).
\end{aligned}$$

$\square$

**Lemma 4.4.** *The Gaussian mechanism with parameter $\sigma$ has mean squared $\ell_2$ error $d\sigma^2$.*

*Proof.* Denote the mechanism by $N_\sigma$. Then by linearity of expectation and the fact that the Gaussian mechanism has independent Gaussian marginals,

$$\begin{aligned}
\mathbb{E}_{y \sim N_\sigma}\left[\|y\|_2^2\right] &= \mathbb{E}\left[\sum_{j=1}^d y_j^2\right] \\
&= d\mathbb{E}_{z \sim N(0,\sigma^2)}\left[z^2\right] \\
&= d\sigma^2
\end{aligned}$$

where the last equality used

$$\mathbb{E}_{z \sim N(0,\sigma^2)}\left[z^2\right] = \text{Var}(z) + \mathbb{E}\left[z\right]^2 = \sigma^2.$$

$\square$

The following result provides evidence that, with reasonable parameters, approximate DP does not yield meaningful utility improvements over pure DP for the Laplace mechanism. A result like this is likely folklore, but we include it here for completeness.

**Lemma A.2.** *The Laplace mechanism with parameter $\sigma \leq 1/\varepsilon$ does not satisfy $(\varepsilon, \delta)$-DP for $\varepsilon < 2\ln(1-\delta) + \frac{1}{\sigma}$.*

*Proof.* Let $T(X) = 0$ and let $T(X') = e_1$. Then $\|T(X) - T(X')\|_1 = 1$, and

$$\mathbb{P}_{y \sim M(0)}\left[\ln\left(\frac{f_X(y)}{f_{X'}(y)}\right) \geq \varepsilon\right] = \mathbb{P}_{y \sim M(0)}\left[\|y - e_1\|_1 - \|y\|_1 \geq \sigma\varepsilon\right] = \mathbb{P}_{y \sim M(0)}\left[|y_1 - 1| - |y_1| \geq \sigma\varepsilon\right].$$

Mechanism $M$ is equivalent to the spherical Laplace distribution where each dimension is drawn from $\text{Lap}(\sigma)$. This distribution has CDF $F(x) = 1 - \frac{1}{2}\exp(-x/\sigma)$ for $x \geq 0$. Condition $|y_1 - 1| - |y_1| \geq \sigma\varepsilon$ holds if and only if $y_1 \leq \frac{1}{2}(1 - \sigma\varepsilon)$, so the probability of drawing such a $y$ is

$$1 - \frac{1}{2}\exp\left(-\frac{1}{\sigma}\left[\frac{1}{2}(1 - \sigma\varepsilon)\right]\right) = 1 - \frac{1}{2}\exp\left(\frac{\varepsilon - \frac{1}{\sigma}}{2}\right)$$

Therefore $\mathbb{P}\left[\ell_{M,X,X'} \geq \varepsilon\right] = 1 - \frac{1}{2}\exp\left(\frac{\varepsilon - \frac{1}{\sigma}}{2}\right)$.

We now analyze $\mathbb{P}\left[\ell_{M,X',X} \leq -\varepsilon\right]$. Because $\ell_{M,X',X} = \log(f_{X'}(y)/f_X(y)) = \frac{1}{\sigma} \cdot (|y_1| - |y_1 - 1|)$, we get

$$\mathbb{P}\left[\ell_{M,X',X} \leq -\varepsilon\right] = \mathbb{P}_{y \sim M(1)}\left[|y_1 - 1| - |y_1| \geq \sigma\varepsilon\right]$$

where $M(1)$ denotes the $d$-dimensional Laplace mechanism centered at $e_1$. By the same logic used above, $|y_1 - 1| - |y_1| \geq \sigma\varepsilon$ if and only if $y_1 \leq \frac{1}{2}(1 - \sigma\varepsilon)$. For $y \sim M(1)$, this event has the same probability as $y_1' \leq \frac{1}{2}(1 - \sigma\varepsilon) - 1 = \frac{1}{2}(-1 - \sigma\varepsilon)$ when $y' \sim M(0)$. Furthermore, $\mathsf{Lap}\,(\sigma)$ has CDF $F(x) = \frac{1}{2}\exp(x/\sigma)$ for $x < 0$. Thus $\mathbb{P}\left[\ell_{M,X',X} \leq -\varepsilon\right] = \frac{1}{2}\exp\left(\frac{1}{2}\left[-\frac{1}{\sigma} - \varepsilon\right]\right)$.

Combining these results and applying $1 + x \leq e^x$ yields

$$\mathbb{P}\left[\ell_{M,X,X'} \geq \varepsilon\right] - e^\varepsilon \mathbb{P}\left[\ell_{M,X',X} \leq -\varepsilon\right] = 1 - \frac{1}{2}\exp\left(\frac{\varepsilon - \frac{1}{\sigma}}{2}\right) - e^\varepsilon \frac{1}{2}\exp\left(\frac{1}{2}\left[-\frac{1}{\sigma} - \varepsilon\right]\right)$$

$$= 1 - \exp\left(\frac{\varepsilon - \frac{1}{\sigma}}{2}\right)$$

By Lemma 2.3, this last quantity must be upper bounded by $\delta$ for $M$ to be $(\varepsilon, \delta)$-DP. Rearranging yields the expression in the claim. $\qquad\square$

