# OpenReview forum: "Approximate Differential Privacy of the $\ell_2$ Mechanism"
_ICML.cc/2025/Conference — ICML 2025 poster_

### Official Review · Reviewer_Vdrp · 2025-03-12

**Overall Recommendation:** 4

**Summary:**

This paper studies the $\ell_2$ mechanism for releasing a $d$-dimensional statistic with bounded $\ell_2$ sensitivity under approximate differential privacy.
To release a $d$-dimensional statistic $T(x)$, the $\ell_2$ mechanism samples an output $f_X(y)\propto \exp(-\lVert y - T(x)\rVert_2 / \sigma)$ for suitable $\sigma$.
This can be compared to the Gaussian mechanism which outputs $f_X(y)\propto \exp(-[\lVert y - T(x)\rVert_2 / \sigma]^2)$, i.e., where the norm comes in squared.
The $\ell_2$ mechanism can be viewed as an instantiation of the $K$-norm mechanism (Hardt & Talwar, 2010), and so it is known to satisfy $\frac{1}{\sigma}$-DP.
Its approximate DP properties, however, are not well understood, and exploring these is the main topic of the paper.

To prove that the $\ell_2$ mechanism supports approximate DP, they use the fact that a mechanism $M$ is $(\varepsilon, \delta)$-DP iff $\Pr[\ell_{M, X, X'} \geq \varepsilon] - e^{\varepsilon}\Pr[\ell_{M, X', X} \leq -\varepsilon] \leq \delta$ where $\ell_{M, X', X}$ is the privacy loss of $M$ on arbitrary neighboring datasets $X, X'$ (Balle & Wang, 2018).
Upper bounding the first term and lower bounding the second in the left-hand side becomes a means to computing a $\delta$ for a given $\epsilon$.
The authors show that both terms can be defined by the probability mass that $M(X)$ and $M(X')$ respectively assigns to regions in $\mathbb{R}^d$ defined by specific spherical caps.
They give algorithms for bounding both terms, which, when combined in Algorithm 3, allows for checking if a $(\varepsilon, \delta)$-guarantee holds for a given $d$ and $\sigma$.

Empirically, they demonstrate that binary searching over $\sigma$ using Algorithm 3 allows for tight privacy analysis of the mechanism (Figure 4).
They also demonstrate that analytically computing the optimal $\sigma$ for the $\ell_2$ mechanism is feasible and does not scale with $d$ (Figure 5), even if it is slower than the corresponding computations for Laplace and Gaussian mechanisms for a $\ell_2$ guarantee.
They further show that sampling the $\ell_2$ mechanism takes roughly only a factor ~2 more time than the Laplace/Gaussian counterpart (Figure 6).
Figure 1 makes the point that the $\ell_2$ mechanism can meaningfully improve on the mean squared error over the Gaussian mechanism for $d$ up to $100$ (for $\varepsilon=1, \delta=10^{-5}$).

## update after rebuttal

The authors answered my questions adequately. I stand by my score.

**Claims And Evidence:**

Yes.

**Essential References Not Discussed:**

Not that I am aware of.

**Experimental Designs Or Analyses:**

Yes, all the listed experiments look sound to me.
I have minor questions regarding the parameter ranges listed later.

**Methods And Evaluation Criteria:**

Yes.

**Other Comments Or Suggestions:**

I think the paper could benefit from a more thorough investigation of when the $\ell_2$ mechanism improves over the Gaussian mechanism. Figure 1 (Left) shows what happens up to $d=100$ for a fix choice of $(\varepsilon, \delta)$, but it would be interesting to see what happens for larger $d$ and other privacy regimes.

**Other Strengths And Weaknesses:**

Strengths:
1. Well-written paper with clearly stated contributions.
2. The privacy analysis of the $\ell_2$ mechanism appears to be tight (Figure 4), and the fact that it can (at least sometimes) beat Gaussian noise in mean squared error is interesting (and potentially practically useful).
3. As demonstrated by Figure 5 and 6, it also appears as if the $\ell_2$ mechanism is scalable with respect to time.

Weaknesses:
1. Some parameter ranges for the experiments could need more motivation, e.g. why $d=100$ everywhere?
2. Connected to the preceding point, the improvement over the Gaussian/Laplace mechanism seems concentrated to moderately small $d$ for reasonable $(\varepsilon, \delta)$.

**Questions For Authors:**

I ask the following questions to better understand the results. I do not expect my score to change dramatically based on answers to these questions in isolation.

Questions:
1. From the discussion in Section 4 regarding Figure 1, it seems as if the mean squared error of the $\ell_2$ mechanism approaches that of the Gaussian as $d$ increases for general settings of $(\varepsilon, \delta)$. Is this true in general, and if so, is there a simple argument for why that is? I think the improvement for small $d$ is interesting in its own right, but it would be useful to know if there is any benefit to using the $\ell_2$ mechanism for larger $d$.
2. Does the computation in Figure 5 meaningfully depend on $(\varepsilon, \delta)$? I could believe that $d$ has no influence, but it is not clear to me if the computation becomes prohibitive for certain regimes of $(\varepsilon, \delta)$.

**Relation To Broader Scientific Literature:**

The paper gives the first useful (in the sense of efficiently computable) approximate DP guarantee for the $\ell_2$ mechanism.
This mechanism is already known as an instantiation of the $K$-norm mechansim (Hardt & Talwar, 2010), but the approximate DP analysis is novel.
In (Ganesh & Zhao, 2021) they also used spherical caps to bound the privacy loss of "generalized Gaussians", but these are (1) different probability distributions and (2) they give looser bounds.

**Theoretical Claims:**

Yes, I read through the proofs in the main part of the paper. I did not check every detail, but it seemed convincing to me. I did not check the appendix however.

---

> ### Author Rebuttal · Authors · 2025-03-31
>
> Thanks for the review!
>
> > Some parameter ranges for the experiments could need more motivation, e.g. why $d=100$ everywhere?
>
> Our experiments focus on the $d \leq 100$ setting to highlight the range where the $\ell_2$ mechanism offers the largest improvement over both Laplace and Gaussian noise. As noted at the end of Section 4.2, the gap between $\ell_2$ and Gaussian noise shrinks to $<1$% at $d=350$ (and essentially vanishes for larger $d$). If desired, we can add more language highlighting this behavior.
>
> > Connected to the preceding point, the improvement over the Gaussian/Laplace mechanism seems concentrated to moderately small $d$ for reasonable $(\varepsilon, \delta)$.
>
> We agree that the $\ell_2$ mechanism's improvement is concentrated to moderate $d$ (e.g., $d < 500$, and most notably on $d < 100$). Regarding the privacy regime, we note that the curve in the left plot of Figure 1 looks essentially the same in higher ($(0.1, 10^{-7})$-DP) and lower ($(10,10^{-3})$-DP) privacy settings, as mentioned at the end of Section 4.2.
>
> > From the discussion in Section 4 regarding Figure 1, it seems as if the mean squared error of the $\ell_2$ mechanism approaches that of the Gaussian as $d$ increases for general settings of $(\varepsilon, \delta)$. Is this true in general, and if so, is there a simple argument for why that is? I think the improvement for small $d$ is interesting in its own right, but it would be useful to know if there is any benefit to using the mechanism for larger $d$.
>
> Our experiments demonstrate that the $\ell_2$ mechanism error converges to Gaussian mechanism error as $d$ increases, irrespective of the $(\varepsilon, \delta)$ regime. However, we do not have a formal proof that this holds. A possible intuitive explanation is that the $(\varepsilon, \delta)$-DP $\ell_2$ mechanism ``spends its $\delta$'' violating $\varepsilon$-DP close to the origin, while the Gaussian mechanism continues doing so arbitrarily far into its tails (and thus can incur arbitrarily high privacy loss). The result is that the $\ell_2$ density peaks at a smaller $\ell_2$ distance than the Gaussian density at the same $(\varepsilon, \delta)$-DP guarantee, but this difference shrinks as $d$ grows. This partially explains why the $\ell_2$ mechanism error approaches the Gaussian mechanism error as $d$ grows, though it does not explain why the chosen $\sigma$s necessary for DP lead to this behavior.
>
> > Does the computation in Figure 5 meaningfully depend on $(\varepsilon, \delta)$? I could believe that $d$ has no influence, but it is not clear to me if the computation becomes prohibitive for certain regimes of $(\varepsilon, \delta)$.
>
> We do not believe that the ($\sigma$) computation in Figure 5 meaningfully depends on the exact privacy parameters within typical ranges. We tested this computation at more extreme privacy levels ($(0.1, 10^{-7})$-DP and $(10,10^{-3})$-DP) and did not encounter numerical issues. However, this might change as $e^\varepsilon$ and $\delta$ near the limits of conventional floating point accuracy.

---

### Official Review · Reviewer_xaZ5 · 2025-03-14

**Overall Recommendation:** 3

**Summary:**

The paper studies the L2 mechanism with bounded L2 sensitivity in d dimensions, demonstrating improvements over the Laplace and Gaussian mechanisms under approximate differential privacy. It presents algorithms for computing approximation bounds for privacy loss random variables and introduces a parallel sampler for generating noise from the resulting distribution.

## update after rebuttal
I will keep my score unchanged.

**Claims And Evidence:**

The approximation bounds for privacy loss random variables are novel, providing new results for achieving differential privacy in d dimensions.

**Essential References Not Discussed:**

No.

**Experimental Designs Or Analyses:**

The experiment shows the proposed mechanism is tight and can be efficient.

**Methods And Evaluation Criteria:**

The experiment support the claims. The sampling is efficient.

**Other Comments Or Suggestions:**

No.

**Other Strengths And Weaknesses:**

One missing aspect is the composition of the L2 mechanism. The Gaussian mechanism is widely used in DP-SGD due to the availability of numerical composition analysis. To better demonstrate the practicality of the proposed L2 mechanism, its composition must be studied.

**Questions For Authors:**

In Figure 1, the L2 mechanism exhibits a similar error to the Gaussian mechanism when d >100. In DP-SGD, where the dimensionality often exceeds 1000, does this imply that the L2 mechanism has comparable error to the Gaussian mechanism? If so, what is the advantage of using the L2 mechanism? It appears that the L2 mechanism achieves lower error for moderate d, but this benefit may not be significant in applications like DP-SGD.

A related concern arises in Figure 6, where the sampling process for the L2 mechanism appears less stable. If d >1000, will the sampling time increase compared to the Gaussian mechanism? If so, this could be a notable limitation, as the additional computational overhead may not be negligible.

**Relation To Broader Scientific Literature:**

The paper reduces the error of dp mechanism in d dimension case.

**Theoretical Claims:**

I checked most of the proofs in the main paper, they make sense to me.

---

> ### Author Rebuttal · Authors · 2025-03-31
>
> Thanks for the review!
>
> > One missing aspect is the composition of the L2 mechanism. The Gaussian mechanism is widely used in DP-SGD due to the availability of numerical composition analysis. To better demonstrate the practicality of the proposed L2 mechanism, its composition must be studied.
>
> We agree that analyzing composition is the logical next step for studying the $\ell_2$ mechanism. A basic approach can use existing generic composition results for pure and approximate DP algorithms, but this is unlikely to obtain better error than the Gaussian mechanism beyond a very small number of compositions. Possible alternative approaches include:
>
> 1) Investigating advanced composition for mechanisms that satisfy simultaneous (and nontrivial) pure and approximate DP guarantees. For example, at $d=50$, at $\sigma \approx 0.5$ the $\ell_2$ mechanism is both $(1, 10^{-5})$-DP and $\approx 2$-DP. We are not aware of advanced composition bounds that can take advantage of both guarantees.
>
> 2) Analyzing the moment generating function of the privacy loss random variable. Such an analysis could provide better privacy accounting using CDP or RDP or FFT-based privacy accounting. As demonstrated in the paper, even deriving tail bounds for the privacy loss distribution is already involved, but it is possible that some kind of moment analysis can be done.
>
> We suggest that results in this direction are a good candidate for future work.
>
> > In Figure 1, the L2 mechanism exhibits a similar error to the Gaussian mechanism when d >100. In DP-SGD, where the dimensionality often exceeds 1000, does this imply that the L2 mechanism has comparable error to the Gaussian mechanism? If so, what is the advantage of using the L2 mechanism? It appears that the L2 mechanism achieves lower error for moderate d, but this benefit may not be significant in applications like DP-SGD.
>
> Yes, our experiments show that the $\ell_2$ mechanism's error converges to the Gaussian mechanism's error as $d$ grows, and at $d=1000$, the gap is essentially 0 (a short discussion of intuition for this effect appears in our response to Reviewer Vdrp). We therefore agree that the $\ell_2$ mechanism does not meaningfully improve over the Gaussian mechanism in very high-dimensional settings. However, we suggest that the moderate-$d$ setting covers many practical problems and is therefore still worth studying. If desired, we can add a short discussion highlighting this guidance for the dimension ranges where $\ell_2$ noise is most useful.
>
> > A related concern arises in Figure 6, where the sampling process for the L2 mechanism appears less stable. If d >1000, will the sampling time increase compared to the Gaussian mechanism? If so, this could be a notable limitation, as the additional computational overhead may not be negligible.
>
> The $\ell_2$ mechanism is a constant factor (in $d$) slower in our implementation. However, we note that both the $\ell_2$ and Gaussian mechanism sampling algorithms are parallelizable, and the parallelized runtime is independent of $d$. The experiments provided here do not attempt this parallelization, which is why their runtime increases with $d$. As described in Section 3.2, in a parallel setting, the $\ell_2$ sampler adds one more map and combine step over the Gaussian sampler. In a parallelized system, we expect that this cost is mild.

---

### Official Review · Reviewer_sZuF · 2025-03-16

**Overall Recommendation:** 1

**Summary:**

The authors consider a specific instantiation of the K-norm mechanism using an L2 norm.  They establish conditions for achieving approximate DP as opposed to pure DP as was done in the original K-norm paper. Theory and experiments are provided.

**Claims And Evidence:**

Yes, the paper provides both theory and simulations to demonstrate the improvement.

**Essential References Not Discussed:**

While the mechanism looks like it is wrapped up in an exponential type mechanism, it can also be viewed as an additive perturbation where $\tilde T(X) = T(X) + Z$ where $Z$ is distributed as an "l2" random variable.  In that sense, there are a lot of options available for pure/approximate DP that can be discussed.  In fact, the distribution used in the paper has been viewed as a multivariate version of the Laplace distribution (there are multiple extensions).

**Experimental Designs Or Analyses:**

Experiments seemed fine.

**Methods And Evaluation Criteria:**

Yes, both theory and simulations are provided to demonstrate the work.

**Other Comments Or Suggestions:**

None.

**Other Strengths And Weaknesses:**

In general the paper is a very narrow contribution as they study a very specific type of DP for a very specific mechanism that doesn't seem to be used much in the literature.  The theory of the paper is also very unpleasant to read.  Major theorems should be organized and presented as the key results.  Especially novel lemmas certainly can as well, but the rest should be in the appendix.

**Questions For Authors:**

None.

**Relation To Broader Scientific Literature:**

The related work section was quite lacking.  4 papers are mentioned in field that has existed for 20 years.  I understand that the authors are working with approximate DP, where as a lot of related methods work on pure, but the authors should still do a better job placing their work within the context of the broader literature.

**Theoretical Claims:**

I checked some of the early lemmas and the results seemed sound.  However, the communication of the theory I found to be unpleasant.  I'm not sure I've ever seen so many lemmas without a single theorem.  The bulk of the paper reads more like an appendix.

---

> ### Author Rebuttal · Authors · 2025-03-30
>
> Thanks for the review!
>
> > I checked some of the early lemmas and the results seemed sound. However, the communication of the theory I found to be unpleasant. I'm not sure I've ever seen so many lemmas without a single theorem. The bulk of the paper reads more like an appendix … [t]he theory of the paper is also very unpleasant to read. Major theorems should be organized and presented as the key results. Especially novel lemmas certainly can as well, but the rest should be in the appendix.
>
> The paper's current presentation attempts to highlight the main conceptual ideas of the privacy analysis in the main body while deferring most of the calculations to the appendix. We chose this presentation because direct approximate DP analyses of additive noise mechanisms are rare in the existing literature, and we believe the form of the analysis is interesting. Since the overall analysis has a single end goal (an approximate DP guarantee), describing the constituent results as lemmas seemed appropriate. If it seems helpful, we can add an overall privacy theorem.
>
> > The related work section was quite lacking. 4 papers are mentioned in field that has existed for 20 years. I understand that the authors are working with approximate DP, where as a lot of related methods work on pure, but the authors should still do a better job placing their work within the context of the broader literature.
>
> To the best of our knowledge, this section discusses all of the existing work on the $\ell_2$ mechanism, and the Laplace and Analytical Gaussian Mechanisms discussed elsewhere are the most relevant baselines. Do you have other references in mind?
>
> > While the mechanism looks like it is wrapped up in an exponential type mechanism, it can also be viewed as an additive perturbation where $\tilde T(X) = T(X) + Z$ where $Z$ is distributed as an "l2" random variable. In that sense, there are a lot of options available for pure/approximate DP that can be discussed.
>
> The suggested additive noise perspective is also valid (and appears in the discussion of the $K$-norm mechanism in Lemma 2.4). We agree that other additive noise mechanisms exist, and as mentioned above, to the best of our knowledge the most common mechanisms in theory and practice for computing an $\ell_2$ sensitive statistic are the variants of Laplace and Gaussian noise featured in the paper (though additional references are welcome!).
>
> > In fact, the distribution used in the paper has been viewed as a multivariate version of the Laplace distribution (there are multiple extensions).
>
> Can you elaborate on this? The multivariate Laplace distribution depends on $\ell_1$ distance to the true statistic, but the $\ell_2$ mechanism depends on $\ell_2$ distance. In what sense is the $\ell_2$ mechanism a multivariate version of the Laplace distribution?
>
> > In general the paper is a very narrow contribution as they study a very specific type of DP for a very specific mechanism that doesn't seem to be used much in the literature.
>
> We agree that the $\ell_2$ mechanism is not currently common in the literature, in part because our paper is the first to prove a (nontrivial) approximate DP guarantee for it. As the results in the paper show, this approximate DP analysis enables the $\ell_2$ mechanism to obtain lower error than the Laplace and Gaussian baselines, which appear in most DP papers. Approximate DP is the most common notion of DP in the literature and in practice. For example, it is the primary definition used in usability studies of DP [1, 2] as well as industry deployments [3, 4, 5]. Can you elaborate on why it might be considered "a very specific type of DP"?
>
> [1] https://arxiv.org/abs/2302.11775
>
> [2] https://arxiv.org/abs/2406.12103
>
> [3] https://journalprivacyconfidentiality.org/index.php/jpc/article/view/782
>
> [4] https://arxiv.org/abs/1909.01917
>
> [5] https://arxiv.org/abs/2201.11603

---

> > ### Comment · Reviewer_sZuF · 2025-04-02
> >
> > I appreciate the comments from the authors.
> > a) Ultimately it is the authors' call.  But I would argue that the purpose of Lemmas is to decompose the steps of establishing a theorem into more manageable pieces.  I think most of the mathematics community would agree with me.  Certainly lemmas can be included in the main body of the paper to help tell the story, but I don't think they are currently acting in that way.
> >
> > b) Why would you only discuss the l2 mechanism when discussing the literature?  I think the work should be placed more broadly within the literature on additive noise mechanisms.
> >
> > c) The multivariate Laplace is not a uniquely defined distribution.  There are multiple ways to generalize the univariate Laplace distribution.  One way is to "preserve the norm" within the density, which would basically yield an l2 or l1 type mechanism.  Another way is to say that a multivariate Laplace must have Laplace marginals, in which case you define the characteristic function in a very specific way.  It is an interesting (though not crucial) connection.
> >
> > d) So the authors claim that for approximate DP, the l2 mechanism adds less noise than the Gaussian -- which theorem shows that?

---

> > > ### Author Response · Authors · 2025-04-03
> > >
> > > Thanks for following up!
> > >
> > > > Ultimately it is the authors' call. But I would argue that the purpose of Lemmas is...
> > >
> > > We'll take another look at the organization of the proof exposition. If you have specific suggestions, please let us know.
> > >
> > > > Why would you only discuss the l2 mechanism when discussing the literature? I think the work should be placed more broadly within the literature on additive noise mechanisms.
> > >
> > > First, we note that the paper discusses both the Laplace and Gaussian baselines in the introduction and experiments. As the literature for DP-SGD, the binary tree mechanism, and the projection, factorization, and matrix mechanism (see Introduction for references) demonstrates, these are the baseline algorithms for privately computing an $\ell_2$ sensitive statistic. The Related Work also discusses generalized Gaussian mechanisms (though ultimately argues that their relevance is because of a similarity in one step of the analysis, rather than utility for the problem in question). We therefore suggest that the current presentation is reasonably sufficient context.
> > >
> > > There are a few more additive noise mechanisms that we considered discussing. In each case, we largely decided against including them because they are less relevant than the Laplace or Gaussian baselines. However, discussing them here may be useful context.
> > >
> > > 1) $K$-norm mechanism. The $K$-norm mechanism is particularly useful when applied to statistics whose sensitivity is not nicely characterized by an $\ell_p$ norm (see for example [1]). In contrast, we focus on an $\ell_2$-sensitive statistic, so discussing instances other than the $\ell_2$ mechanism doesn't seem relevant. Note that the paper does discuss the $K$-norm mechanism as a generalization of the $\ell_2$ mechanism.
> > >
> > > 2) Staircase mechanism [2]. This mechanism dominates the Laplace mechanism under pure DP. However, it is only noticeably better than the Laplace mechanism in the very low-privacy/large $\varepsilon$ setting (see Figure 2 in [2] – note also that this figure is for the 1-dimensional staircase mechanism; a $d$-dimensional staircase mechanism would need an even larger $\varepsilon$ to separate from the Laplace mechanism, because it needs a large $\varepsilon$ in each coordinate).
> > >
> > > 3) Bounded noise mechanisms [3]. These mechanisms were developed for computing a private statistic with bounded $\ell_\infty$ sensitivity, and the paper provides an $(\varepsilon, \delta)$-DP algorithm that in some settings beats the Gaussian mechanism for this problem. However, the primary mechanism studied in the paper has several drawbacks: to the best of our knowledge, no algorithm to sample the mechanism is known, and it only obtains lower error than the Gaussian mechanism when $k$ is very large (approximately $> 1000$) and a very high probability ($\gg 0.99$) $\ell_\infty$ error bound is desired (see Figure 1 in [3]). Since we focus on practical algorithms for $\ell_2$ sensitive statistics, this did not seem relevant.
> > >
> > > In summary, the paper discusses all of the additive noise mechanisms that, to the best of our knowledge, are most relevant to the problem at hand. If you have additional concrete examples in mind, please let us know!
> > >
> > > [1] https://arxiv.org/abs/2309.15790
> > >
> > > [2] https://arxiv.org/abs/1212.1186
> > >
> > > [3] https://arxiv.org/abs/2012.03817
> > >
> > > > The multivariate Laplace is not a uniquely defined distribution...
> > >
> > > If we understand correctly, the observation here is that there is no canonical definition for the multivariate Laplace distribution, and a possible definition coincides with the $\ell_2$ mechanism. That seems reasonable. In our experience "multivariate Laplace" typically refers to the Laplace marginals (i.e., $\ell_1$ mechanism) interpretation in the context of DP, which is why we chose the name $\ell_2$ mechanism.
> > >
> > > > So the authors claim that for approximate DP, the l2 mechanism adds less noise than the Gaussian -- which theorem shows that?
> > >
> > > The evidence for our claim is empirical: we prove that our algorithm returns a $(\varepsilon, \delta)$-DP mechanism, and then show experimentally that the returned mechanisms are more accurate than the Analytical Gaussian Mechanism (AGM), particularly when $d$ is not too large. The current presentation tries to be explicit about this (e.g., the claim that it "empirically dominates…" in the Introduction).
> > >
> > > We agree that the best possible version of this paper would include a formal utility result. (Informally, the main obstacle is that the expression for the cap fraction given in Lemma 3.9 has no closed form. This cap fraction is what characterizes the high privacy loss region, so reasoning about the overall accuracy is analytically difficult; the AGM has a similar dependence on the standard normal CDF.) Nonetheless, we suggest that an efficient and provably DP algorithm that obtains clear and nontrivial empirical error improvements for something as widely used as $\ell_2$ sensitive statistics is interesting even without this formal result.

---

### Decision · Program_Chairs · 2025-05-01

**Decision:**

Accept (poster)

**Comment:**

In this paper, the authors give the first approximate differential privacy analysis for a version of the K-norm mechanism in which the norm is the standard Euclidean one. They also empirically show the mechanism is adds less error than the Gaussian noise mechanism for moderate dimensions, but this advantage disappears as the dimension increases. This somewhat limits the impact of the results. Another limiting factor is the lack of tight composition guarantees - fairly tight composition analyses of the Gaussian noise mechanism through tools like Gaussian differential privacy are available, and important for applications like private stochastic gradient descent. Nevertheless, this is an interesting contribution and can be accepted if there is space in the program.